# Redosing with Intralymphatic GAD-Alum in the Treatment of Type 1 Diabetes: The DIAGNODE-B Pilot Trial

**DOI:** 10.3390/ijms26010374

**Published:** 2025-01-04

**Authors:** Rosaura Casas, Andrea Tompa, Karin Åkesson, Pedro F. Teixeira, Anton Lindqvist, Johnny Ludvigsson

**Affiliations:** 1Division of Pediatrics, Department of Biomedical and Clinical Sciences, Faculty of Medicine and Health Sciences, Linköping University, 581 83 Linköping, Sweden; rosaura.casas@liu.se; 2Department of Clinical Diagnostics, School of Health and Welfare, Jönköping University, 551 11 Jönköping, Sweden; andrea.tompa@ju.se; 3Division of Medical Diagnostics, Department of Laboratory Medicine, Ryhov County Hospital, 551 85 Jönköping, Sweden; 4Department of Pediatrics, Ryhov County Hospital, 551 85 Jönköping, Sweden; karin.akesson@liu.se; 5Diamyd Medical AB, 111 56 Stockholm, Sweden; pedro.teixeira@diamyd.com (P.F.T.); anton.lindqvist@diamyd.com (A.L.); 6Crown Princess Victoria Children’s Hospital, Linköping University, 581 85 Linköping, Sweden

**Keywords:** autoantigen, immunotherapy, GAD-alum, lymph node, intralymphatic, type 1 diabetes, redosing

## Abstract

Immunotherapies aimed at preserving residual beta cell function in type 1 diabetes have been successful, although the effect has been limited, or raised safety concerns. Transient effects often observed may necessitate redosing to prolong the effect, although this is not always feasible or safe. Treatment with intralymphatic GAD-alum has been shown to be tolerable and safe in persons with type 1 diabetes and has shown significant efficacy to preserve C-peptide with associated clinical benefit in individuals with the human leukocyte antigen DR3DQ2 haplotype. To further explore the feasibility and advantages of redosing with intralymphatic GAD-alum, six participants who had previously received active treatment with intralymphatic GAD-alum and carried HLA DR3-DQ2 received one additional intralymphatic dose of 4 μg GAD-alum in the pilot trial DIAGNODE-B. The participants also received 2000 U/day vitamin D (Calciferol) supplementation for two months, starting one month prior to the GAD-alum injection. During the 12-month follow-up, residual beta cell function was estimated with Mixed-Meal Tolerance Tests, and clinical and immune responses were observed. C-peptide decreased minimally, and most patients showed stable HbA1c and IDAA1c. The mean % TIR increased while the mean daily insulin dose decreased at month 12 compared to the baseline. Redosing with GAD-alum seems to be safe and tolerable, and may prolong the disease modification elicited by the original GAD-alum treatment.

## 1. Introduction

The incidence of type 1 diabetes (T1D) is increasing worldwide for reasons that remain unknown. Despite recent promising advances with immune and cell therapies, preventing stage 3 is difficult, and there is currently no cure. Even though treatment of T1D has improved, with better insulin analogues, glucose sensors, and “smarter” pumps, the disease management burden is heavy. Most patients do not reach treatment targets [1,2] and often suffer from hypoglycemia and anxiety, long-term disease complications resulting in decreased life expectancy [3,4,5,6]. Interventions to preserve residual beta cell function aimed at the immune system [7] have been tried since the 1970s [7,8,9,10,11,12,13,14,15,16,17], but their effect has been limited and transient, with some interventions raising safety concerns [13,18,19,20,21,22,23,24,25,26,27,28,29,30,31,32,33,34,35,36,37,38,39,40,41,42]. Transient effects may require redosing, but administration of an extra dose may not always be feasible or safe [43].

The rapid progress in the last few years has created hope [44], with new efficacious treatments including the first Phase III trial with teplizumab showing significant preservation of beta cell function [37,45]. However, although teplizumab is clinically available in the USA for high-risk individuals in stage 2 [45], this treatment has not yet been approved for treatment of newly diagnosed T1D. In order for immune interventions to affect the quality of life of individuals living with T1D, such therapies must be not only efficacious but also safe, and not add to the already heavy burden of disease and disease management.

Antigen-specific immunotherapy with glutamic acid decarboxylase (GAD)_65_ formulated with aluminum hydroxide (GAD-alum) aims to modulate the autoimmune response to the beta cell antigen GAD65 thereby interrupting the disease progression. The treatment has been shown to be easy and safe to administer, but the clinical results were inconclusive [46,47,48], even though meta-analysis of the data supported a high probability that the treatment was efficacious [49]. To improve the effect of GAD-alum, the drug was injected into an inguinal lymph node in the open-label clinical trial DIAGNODE-1, testing three injections of 4 μg of GAD-alum one month apart. DIAGNODE-1 was the first human trial with an autoantigen given intralymphatically, and the results were positive [50,51,52]. DIAGNODE-1 was followed by a randomized double-blind placebo-controlled Phase IIb trial, DIAGNODE-2. In DIAGNODE-2, significant efficacy in preserving C-peptide was limited to patients with the human leukocyte antigen (HLA) DR3DQ2 haplotype, in line with previous findings in clinical trials [53,54]. The DR and DQ HLA genes code for class II Major Histocompatibility Complex proteins that present antigen peptides to CD4 T-cells, and DR3DQ2 haplotype alleles, are associated with increased risk of developing T1D. To investigate if treatment effects could be extended, a fourth dose was given 2.5 years after the initial treatment to three adult patients with DR3DQ2 in a DIAGNODE-1 extension study, which seemed to prolong the preservation of beta cell function [55].

In addition to the intralymphatic GAD-alum injections, patients have received oral vitamin D, as adequate vitamin D concentrations may play a role in achieving an appropriate effect on the immune system [56]. Vitamin D is thought to improve dendritic cell function and may also induce TH2 deviation [56,57] and a shift of T cells from an effector to a regulatory phenotype [58]. It has also been suggested that vitamin D may protect β cells and improve insulin sensitivity [59], and epidemiological studies have suggested a correlation with type 1 diabetes incidence [60]. However, several studies have contradicted that vitamin D by itself has an effect on preservation of beta cell function [61,62,63,64], in accordance with our own previous results [54]. We therefore in this pilot study focused our investigations on the consequences of redosing with intralymphatic GAD-alum.

It has been shown that administration of GAD-alum both subcutaneously and into the lymph nodes has a specific immunomodulatory effect, indicated by in vitro cytokine secretion and proliferation upon GAD_65_-stimulation [51,52,65,66,67,68] and induction of anti-GAD65 antibodies (GADA) [69]. Furthermore, low doses of GAD-alum directly into the lymph node have had a more potent impact on immune responses, shown by induction of higher GADA titers and GAD-induced cytokine secretion than observed after subcutaneous administration [70].

The aim of this study was to further explore the feasibility and advantages of redosing with intralymphatic GAD-alum participants who previously received active treatment with GAD-alum in DIAGNODE-1 or DIAGNODE-2, and carried HLA DR3-DQ2, received one additional intralymphatic injection and were followed for 12 months in the DIAGNODE-B first-in-human pilot trial

## 2. Results

### 2.1. Safety and Feasibility of Repeated Dosing with GAD-Alum

The design of the DIAGNODE-B clinical trial is illustrated in Figure 1. Briefly, two patients (A1 and 2) received the first injection of GAD-alum (4 µg) into the lymph nodes, followed by two booster injections one month apart in the DIAGNODE-1 open label trial. They received a further dose of GAD-alum (4 µg) after 29 months as an extension of DIAGNODE-1, and a fifth GAD-alum injection (4 µg) 44 months later during the current trial. The other four patients (B1–B4) participated in the placebo-controlled DIAGNODE-2 trial, where they received the first injection of GAD-alum (4 µg) into the lymph nodes, followed by two injections one month apart. They received a fourth dose of GAD-alum (4 µg) during the current trial, patients B1 and B2 after 50 months and patients B3 and B4 after 39 months. All patients took vitamin D (Calciferol) for 120 days during the DIAGNODE-1 and DIAGNODE-2 studies, and a second course of vitamin D (60 days) at the time of the four and fifth injections.

The baseline characteristics of the patients are shown in Table 1. All six enrolled patients completed the trial, and none of the safety assessments raised any concerns. There were no differences in the safety and tolerability between the two patients from the DIAGNODE-1 cohort (A1, A2) who received a fifth injection and the four patients from the DIAGNODE-2 trial (B1, B2, B3, B4) after the fourth dose. Five patients reported 15 treatment-emergent adverse events (TEAEs), nine of them classified as mild and six as moderate events. No serious adverse events were reported. Three patients reported one TEAE each (all transient injection site reactions) assessed by the investigator as related to the study drug. No treatment-emergent episodes of severe hypoglycemia or diabetic ketoacidosis were reported.

### 2.2. Clinical Response

Only small decreases in endogenous insulin secretion were observed during the trial (Figure 2; Appendix A). The mean C-peptide AUC (area under the curve) after the Mixed-Meal Tolerance test (MMTT, 0–120 min) decreased from 0.083 nmol/L/min at baseline to 0.053 nmol/L/min and 0.050 nmol/L/min at months 6 and 12, respectively. This corresponded to a mean change ratio of 0.635 from baseline to month 6 and 0.607 to month 12. The absolute decrease in C-peptide AUC response over the course of the trial was low in all patients.

The mean HbA1c serum levels were similar both at six (53.0 mmol/mol) and twelve (55.3 mmol/mol) months as compared to baseline (53.0 mmol/mol) (Appendix A). The mean daily exogenous insulin consumption decreased from 0.701 IU/kg per day at baseline to 0.529 IU/kg per day at month 6 and 0.553 IU/kg per day at month 12 (Appendix A). This corresponded to mean changes from baseline at month 6 of −0.172 IU/kg per day and 0.147 IU/kg per day at month 12.

The mean percentage of time spent in the glycemic target range (TIR) of 3.9 to 10 mmol/L (70 to 180 mg/dL) was similar at baseline (60.7%) and month 6 (58.3%) but increased at month 12 (68.6%) (Figure 3). This corresponded to a mean change of 7.9% from baseline to month 12 (12.9 h). The increase in % TIR was reflected in a decreased percentage of time spent in the hyperglycemic range of >10 mmol/L (>180 mg/dL), with a mean change from baseline to month 12 of −7.7% (26.9 h) (Appendix A).

### 2.3. Individual Longitudinal Follow-Up from Original Treatment

Individual longitudinal data from the DIAGNODE-B trial were combined with results from the preceding trials (DIAGNODE-1 and DIAGNODE-2) for the six patients (Figure 4). The disease progression was clearly seen in the decline in stimulated C-peptide over time after diagnosis (Figure 4a). Beta cell function declined in a linear fashion, with patients A1, A2 and B1 declining more slowly at a similar rate, while patients B2, B3 and B4 declined more rapidly at a similar rate, leveling out as C-peptide levels approached 0.2 nmol/L. Patients A1 and A2 most clearly showed a perturbation of disease development after initial treatment and/or redosing. All patients had a slightly lower rate of decline after dosing, though this was clearly confounded by the expectation that the decline would naturally decrease over time.

As expected, initial HbA1c levels were higher in patients initially randomized closer to diagnosis (Figure 4b, second row). HbA1c declined initially for all patients and most rapidly for those randomized closer to diagnosis. HbA1c increased over time from all patients to a varying degree, from minimum values in the first year after randomization.

The rise in exogenous insulin dose requirements over time mirrors the decrease in C-peptide decline for all patients, including the perturbations observed in connection with dosing (Figure 4c, third row). The latter may also be affected by inclusion in new clinical trial activities.

### 2.4. Immunological Responses

Generally, the results reflect varied immunological responses at different timepoints. However, the statistical analysis did not show significant differences in the immunological responses.

The serum GADA titers varied substantially between the patients at the trial baseline (0 month) (Figure 5a), and the mean GADA titers increased from 92,892 U/mL at baseline to 145,920 U/mL at month 3, 122,630 U/mL at month 6 and 113,408 U/mL at month 12. The mean GAD_65_-induced proliferation of PBMCs increased from 3.5 at baseline to 6.1 at month 6 and 5.4 at month 12 (Figure 5b; Appendix A). Both GADA levels and proliferation SI were low at the time of the first injection (BL) (Figure 5a,b). A transient rise in both GADA and the proliferation of varying amplitudes was observed after each dosing in all individual patients, with occasional exceptions (Figure 4).

Analysis of GAD-induced cytokine secretion upon in vitro stimulation of PBMCs showed an elevation for several cytokines in most of the patients (Appendix A). Analysis of the relative cytokine contribution for each individual when including cytokines that were also analyzed in previous studies (IL-2, -5, -10, -13, -17, IFN-γ and TNF-α) illustrates that redosing with GAD-alum predominantly induced a relative increase in IL-13 (0 m to 12 m) (Figure 5c). While the relative cytokine contribution before the original treatment (BL) differed between patients, IL-13 predominated at the time of the redosing (0 m), i.e., 39–49 months after the latest dosing. Patients A1 and A2 had previously been redosed 30 months after the initial treatment, at which time (30 months) the cytokine profile was similar to that at 0 months (baseline in DIAGNODE-B trial). There was a noticeable contribution of TNF-α to the relative cytokine profile, especially before initial treatment and redosing, which is most likely explained by the high levels characteristic of this cytokine (Appendix A).

## 3. Discussion

The results of the trial suggest that redosing with intralymphatic GAD-alum was tolerable and seemed to be safe. Redosing, even several years after the first course of the therapy, showed the same pattern of immunological response seen at initial treatment in previous trials.

As the DIAGNODE-B trial included only six patients with no control arm, an important limitation of the study, few conclusions can and should be drawn regarding efficacy. Additionally, three out of six individuals had minimal beta cell function at redosing, as indicated by stimulated C-peptide levels, limiting the possibility of seeing effects on disease progression. That said, C-peptide decreased minimally during the 12-month trial period for all patients, in contrast to the expected natural course (7). Regarding clinical outcomes, most patients showed stable and well-controlled blood sugar levels as measured by HbA1c and IDAA1c. The mean % TIR (3.9 to 10 mmol/L) increased, while mean time in hyperglycemic range decreased, and mean daily exogenous insulin consumption decreased at month 12 when compared to baseline. Treatment with GAD-alum affects the immune system and may have most effect in patients with adequate vitamin D concentrations [72], although vitamin D on its own is insufficient for preserving beta cell function.

The lengthy continuous follow-up of individual stimulated C-peptide levels suggests that the six patients represent two different disease trajectories, with patients A1, A2 and B1 showing a slower progression, either as a result of disease heterogeneity or a difference in ability to respond to the treatment. When comparing trajectories against demographics and baseline values, the pattern was not straightforward. It has been previously shown that patients positive for DR3-DQ2 phenotype and negative for DR4-DQ8 possibly respond even better to GAD-alum treatment [53]. Patients A1 and A2 belong to this HLA category, while B1 does not, and nor do the other three patients. One common feature for patients A1, A2 and B1 was that they were initially treated closer to diagnosis than B2 and B3, suggesting that treatment closer to diagnosis might be advantageous. However, this does not apply to subject B4, who was also treated within 90 days of diagnosis. Regarding trajectory in relation to age, while individuals A1 and A2 were above 20 years old at diagnosis, commonly associated with slower disease progression, B1 was only 13 years old. Additionally, despite showing a faster decline, B4 was also above 20 years of age at diagnosis.

Though there was no control group included in the DIAGNODE-B trial, the longitudinal C-peptide values suggest clinical effect based on changes to the disease trajectories. Ideally, an efficacious treatment would be indicated by a break in the trajectory after redosing, or otherwise by a bump or shift in the trajectory indicating a transient delay. No conclusions can be drawn from the initial three-dose treatment as there was no earlier C-peptide data. Although no strong breaks in the trajectory for any subject were observed, there are indications of bumps at both redosing instances for A1 and A2, and possibly for B1 and B4. The perturbation in the trajectory appears largest in the first redosing of A1 and A2, perhaps because the time since the previous treatment administration was shorter.

Due to the small number of patients, furthermore with two different treatment regimens, statistical analysis of any immune markers included in this trial in relation to stimulated C-peptide was precluded. Graphical illustration of the relative cytokine profile showed more pronounced GAD_65_-stimulated IL-13 secretion after GAD-alum redosing in all patients. In the DIAGNODE-1 trial, it was observed that the elevation of cytokine secretion induced by GAD_65_ treatment waned after 30 months, when almost all analyzed cytokines were undetectable with the exception of IL-13, which was elevated compared to both baseline and 15 months [55]. In the DIAGNODE-2 trial, it was shown that GAD_65_-induced secretion of IL-13, IL-10 and IL-5 and proliferation was increased significantly more in the DR3-DQ2 responder group than in non DR3-DQ2 patients [54,73] These factors were also significantly increased in those patients who showed the lowest rate of disease progression after active treatment regardless of HLA [73]. Interleukin-13 has potent anti-inflammatory activities both in vitro and in vivo [74,75,76] and together with IL-5 has a central role in immune regulation and differentiation, governing the onset of inflammatory responses and providing signals that help turn off chronic inflammation and protect tissues from ongoing damage [76,77]. Thus, it is possible that the effect of GAD-alum might be related to the generation of a predominant anti-autoimmune response to GAD_65_ that counteracts proinflammatory factors and generates an environment where autoreactive Th1 effector cells are suppressed, thereby restoring immunological balance.

In conclusion, this DIAGNODE-B pilot trial provides evidence that redosing with GAD-alum seems to be safe and tolerable. Additionally, though there is considerable individual variation, redosing appears to induce an immunological response comparable to original treatment consistent with GAD_65_-specific anti-inflammatory modulation. There was some indication that redosing may prolong the disease modification elicited by the original GAD-alum treatment, but further placebo-controlled studies are needed to confirm this hypothesis and to establish the best redosing interval. If for no other reason, earlier redosing may be preferred to further slow the progression as soon as possible. It cannot be excluded that different autoantigens should be combined depending on endotype of T1D [78,79,80,81,82], and other treatment modalities may need to be added [83,84,85]. As all treatments to preserve residual beta cell function so far have only shown a transient effect, the possibility to redose is crucial.

## 4. Materials and Methods

### 4.1. Trial Design and Participants

DIAGNODE-B (Eudra CT number 2021-005441-32; NCT NCT05351879) was an open-label Phase I/II pilot trial in GADA-positive T1D patients carrying the HLA DR3-DQ2 haplotype who had previously been treated with four intralymphatic injections of GAD-alum within the DIAGNODE-1 trial or three intralymphatic injections of GAD-alum within the DIAGNODE-2 trial.

The DIAGNODE-B trial comprised five visits to the clinic:

Visit 1: Month 0. Screening and informed consent

Visit 2: Month 1, baseline. Mixed-Meal Tolerance Test (MMTT) and intralymphatic administration of GAD-alum

Visit 3: Month 3. Safety follow-up

Visit 4: Month 6. Safety follow-up and MMTT

Visit 5: Month 12. Safety follow-up and MMTT

All patients were evaluated at day 1 (0 months, baseline) and at 1, 3, 6 and 12 months with clinical examination and blood samples. Mixed-Meal Tolerance Tests (MMTT) [71] were performed at day 1 (baseline) and at 6 and 12 months to measure C-peptide (mean area under the curve [AUC]).

Eight patients from two clinics in Sweden were initially invited to participate. One was unable to participate because of pregnancy and one other for practical reasons. Two individuals who had earlier participated in the DIAGNODE-1 extension trial [55] received a fifth injection of 4 μg of GAD-alum (Diamyd Medical, Stockholm, Sweden) into an inguinal lymph node administered using an ultrasound technique, and three individuals who had participated in DIAGNODE-2 [54] received a fourth injection of GAD-alum into an inguinal lymph node. All participants also received 2000 U/day vitamin D (Calciferol) supplementation (Divisun, Meda AB, Solna, Sweden) for 2 months, starting 1 month prior to the GAD-alum injection. Informed consent was obtained from all patients prior to any trial procedures. Safety, beta cell function, diabetes status and immunological responses were followed for 12 months post-baseline.

### 4.2. Investigational Product and Mode of Administration

The GAD-alum (Diamyd^®^) drug product was composed of the rhGAD65 protein formulated in a sterile, non-pyrogenic phosphate-buffered saline containing an aluminum hydroxide adjuvant, Alhydrogel^®^ (Croda Pharma, Fredrikssund, Denmark). GAD-alum (Diamyd^®^; 4 μg, Protein Sciences Corporation, Meriden, CT, USA) was administered as an ultrasound-guided 0.1 mL intralymphatic injection into an inguinal lymph node.

The trial was approved by the relevant regulatory authorities and research ethics board. All participants and their parents/caregivers gave their consent after receiving oral and written information.

### 4.3. Laboratory Tests

Laboratory analyses were performed at Linköping University, Sweden. Blood and serum samples from all the participants were collected at start of trial (0 months), before the GAD-alum redosing (1 months) and after 3, 6 and 12 months. Samples were drawn during the morning hours, and peripheral blood mononuclear cells (PBMCs) were isolated within 24 h using Leucosep (Greiner Bio One, Kremsmünster, Austria) according to the manufacturer’s instructions.

Analysis of serum C-peptide was performed using a solid phase-two side enzyme immunoassay (Mercodia, Uppsala, Sweden). Results for each assay were validated with the inclusion of a human diabetes antigen control (low/high) (Mercodia). The assay is calibrated against the international reference reagent for C-peptide IRR c-peptide 84/510. Inter- and intra-assay variations were 6.6% and 3.5%, respectively.

### 4.4. Serum Antibodies

Serum GAD autoantibodies (GADA) were estimated in duplicate samples by means of a radio-binding assay, using 35S-labeled recombinant human GAD_65_ (rhGAD_65_) as previously described [69]. Sepharose protein A was used to separate free from antibody-bound labelled GAD_65_. Results were expressed as units/mL.

### 4.5. Lymphocyte Proliferation Assay

Proliferative responses were analyzed in PBMCs cultured in triplicate in medium alone (AIM-V medium with β-mercaptoethanol), in the presence of 5 μg/mL rhGAD_65_ (Diamyd Medical, Stockholm, Sweden) or with CD3/CD28 beads (Gibco, Life Technologies AS, Oslo, Norway). After 3 days, cells were incubated with 0.2 µCi of [^3^H] thymidine/well for 18 h and then harvested. The [^3^H] thymidine incorporation was counted in a beta-scintillation counter (Perkin Elmer, Shelton, CT, USA) as counts per minute (cpm). The proliferation was expressed as a stimulation index (SI), calculated as the mean cpm of cells cultured in triplicate in the presence of stimulus divided by the mean cpm of cells with medium alone.

### 4.6. Cytokine Secretion Assay

PBMCs were cultured for 7 days in the presence of 5 μg/mL rhGAD_65_ or in medium alone at 37 °C in 5% CO_2_, as previously described [68]. The cytokines IL-1β, IL-2, IL-4, IL-5, IL-6, IL-7, IL-8, IL-10, IL-12, IL-13, IL-17, interferon (IFN)-γ, tumor necrosis factor (TNF)-α, and chemokines: monocyte chemoattractant protein (MCP)-1, granulocyte (G) colony stimulating factor (CSF), macrophage inflammatory proteins (MIP)-1β, and granulocyte–monocyte (GM)-CSF were detected by Luminex in supernatants collected after 7 days culture. GAD_65_-induced cytokine secretion is given as pg/mL after subtraction of spontaneous secretion.

The cytokines were measured in cell culture supernatants using a Bio-Plex Pro Cytokine Panel (Bio-Rad, Hercules, CA, USA) according to the manufacturer’s instructions. Data were collected using the Luminex 200 ™ (Luminex xMAP™ Corporation, Austin, TX, USA). Antigen-induced cytokine secretion level was calculated by subtracting the spontaneous secretion (i.e., secretion from PBMCs cultured in medium alone) from the one following stimulation with GAD_65_, and the levels were expressed as pg/mL.

### 4.7. Statistical Analysis

Since the sample size was limited, non-parametric tests were used to compare immunological observations at the different timepoints. Friedman’s test was used to compare differences in the immunological responses at the different timepoints. Differences were considered statistically significant for *p* < 0.05.

## Figures and Tables

**Figure 1 ijms-26-00374-f001:**
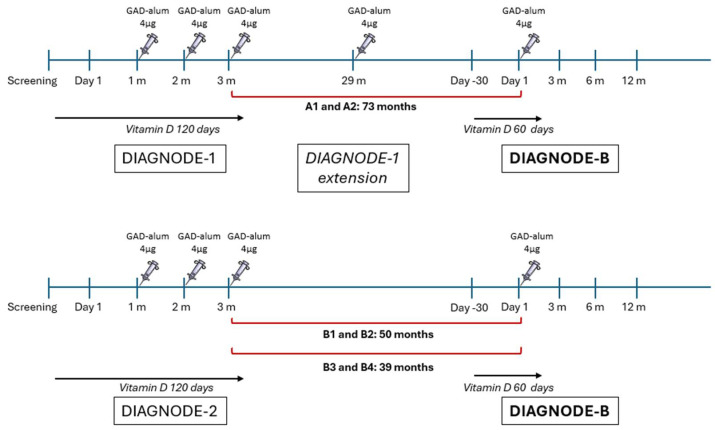
Overview of the GAD-alum treatment in DIAGNODE studies. Patients A1 and A2 (*n* = 2) received a course of three injections of GAD-alum (4 µg) into the lymph nodes (LN) one month apart in the DIAGNODE-1 trial. These patients received a further dose of GAD-alum (4 µg) after 29 months as an extension of DIAGNODE-1, and a fifth GAD-alum injection (4 µg) 44 months later during DIAGNODE-B. Patients who participated in the placebo-controlled DIAGNODE-2 (*n* = 4, B1–B4) trial received a course of three injections of GAD-alum (4 µg) into the lymph nodes (LN) one month apart. A fourth dose of GAD-alum (4 µg) was administered during the current trial, in the case of patients B1 and B2 after 50 months and patients B3 and B4 after 39 months. All patients were also taking vitamin D (Calciferol) from day 1, for 120 days, both in DIAGNODE-1 and DIAGNODE-2. A second course of vitamin D (60 days) was taken by the patients at the time of the four and fifth injections.

**Figure 2 ijms-26-00374-f002:**
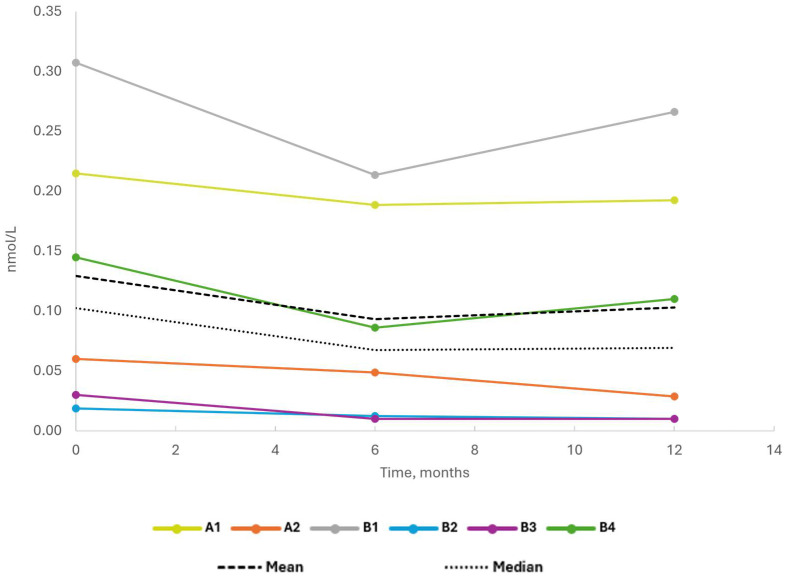
C-peptide AUC. Individual, mean and median C-peptide AUC (area under the curve) after Mixed-Meal Tolerance Test (MMTT) [71] at day 1 (baseline) and at 6 and 12 months. C-peptide is expressed as nmol/L.

**Figure 3 ijms-26-00374-f003:**
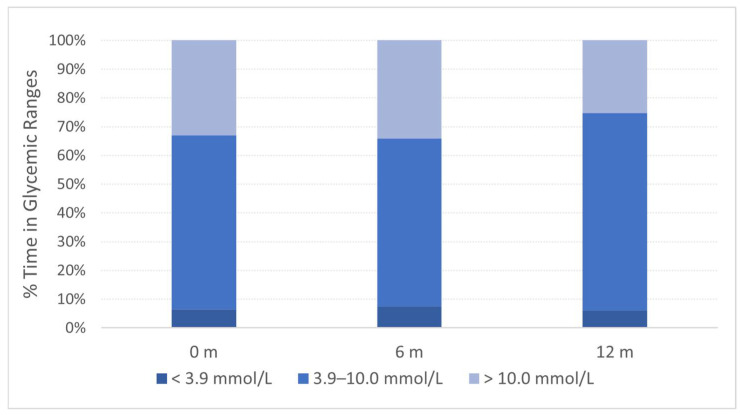
Time in glycemic ranges. Mean percentage of time spent in hypoglycemic (<3.9 mmol/L), glycemic (3.9–10.0 mmol/L) and hyperglycemic (>10.0 mmol/L) range from baseline, 0 months (m), to 12 months (m) follow-up in patients (A1, A2, B1, B2, B3, B4) in the DIAGNODE-B trial.

**Figure 4 ijms-26-00374-f004:**
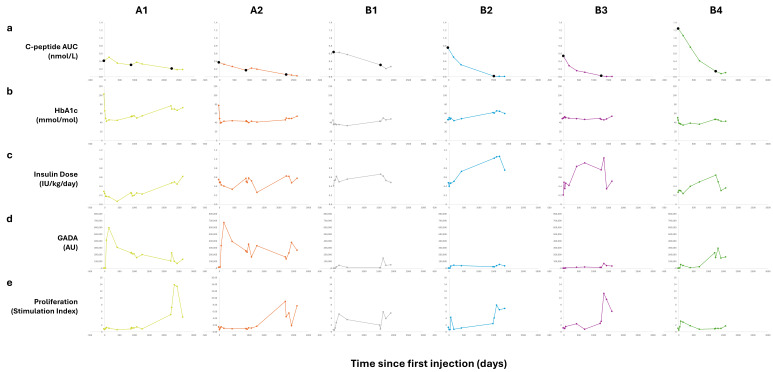
Individual longitudinal follow-up. Individual longitudinal values for: (**a**) C-peptide AUC (first row), (**b**) HbA1c (second row), (**c**) exogenous insulin dose (third row), (**d**) GADA (fourth row) and (**e**) GAD-stimulated proliferation of PBMCs (fifth row) from first GAD-alum treatment to 12 months follow-up in DIAGNODE-B for the patients from the DIAGNODE-1 cohort (patients: A1, A2) and the four patients from the DIAGNODE-2 cohort (patients: B1, B2, B3, B4). Black dots in individual C-peptide plots (first row) mark the instances of GAD-alum injections. The figure combines data from DIAGNODE-1, DIAGNODE-2 and DIAGNODE-B.

**Figure 5 ijms-26-00374-f005:**
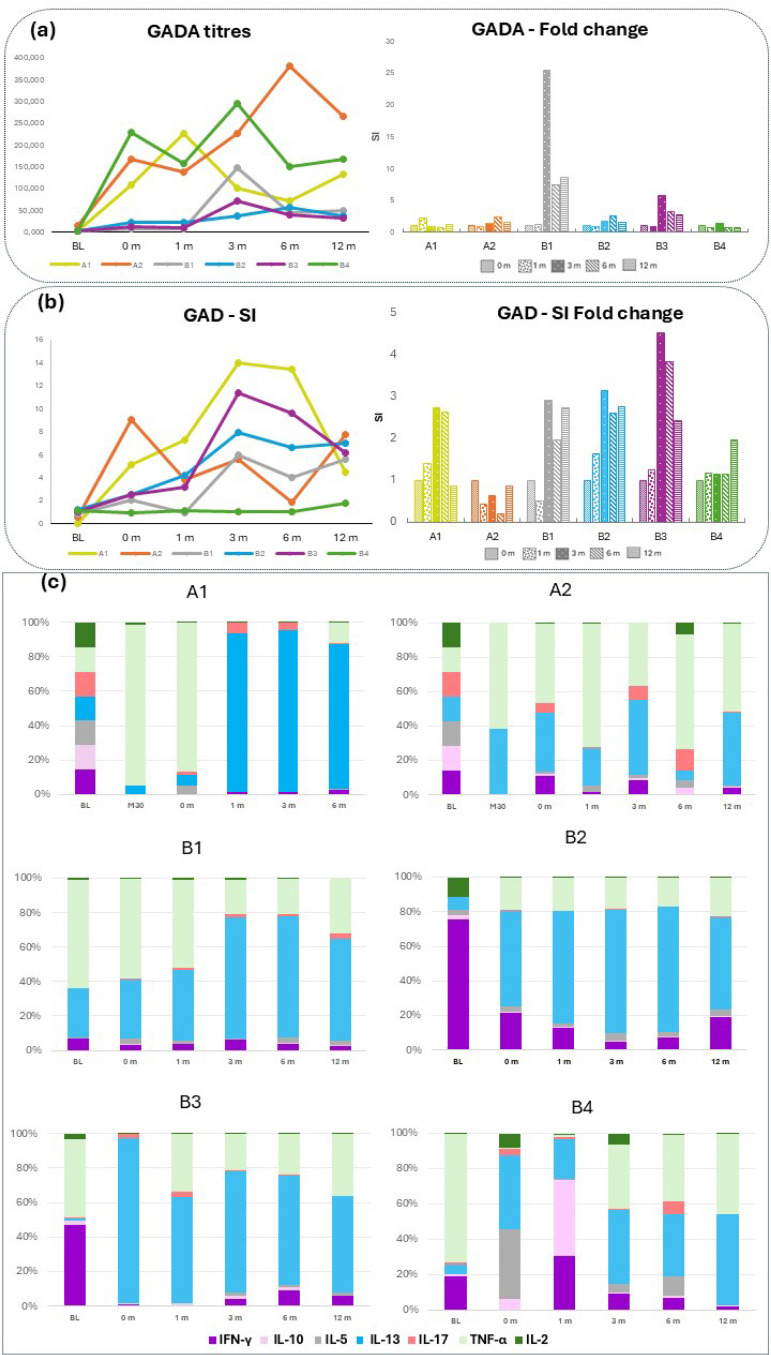
Immunological responses upon in vitro peripheral mononuclear blood cell (PMBC) stimulation in patients (A1, A2, B1, B2, B3, B4) who received an additional injection of GAD-alum in the DIAGNODE-B trial. (**a**) GADA titers (U/mL) expressed as arbitrary units (AUs). Data from DIAGNODE-B. (**b**) PBMC proliferative responses to GAD65. Cells were cultured for 3 days with GAD_65_ (5 µg/mL) or medium; thereafter, cells were pulsed with [3H] thymidine and harvested. Proliferation is expressed as stimulation index (SI) and calculated from the mean of triplicates in the presence of stimulus divided by the mean of triplicates with medium alone. SI scale started from 1 after dividing by the unstimulated value index (SI unstimulated  =  1). (**a**) GADA and (**b**) proliferation fold change were calculated in relation to baseline (BL) values. Data from DIAGNODE-B. (**c**) Cytokine relative contribution (%). Interleukin (IL)-2, -5, -10, -13, -17, IFN-γ and TNF-α were detected by Luminex in PBMCs supernatants after 7-day culture in the presence of GAD_65_ (5 µg/mL) or medium alone. Secretion levels were calculated after subtraction of spontaneous secretion from each individual and expressed as pg/mL D-PBMC proliferative responses to GAD_65_. The results are based on samples collected before the original treatment (BL), before the first redosing 30 months after the initial treatment (30M, only A1 and A2), at start of DIAGNODE-B (0 m), before the GAD-alum redosing (1 m) and after 3, 6 and 12 months.

**Table 1 ijms-26-00374-t001:** Demographics and baseline characteristics. Subject level demographic data and baseline characteristics both at the time of first treatment with GAD65 and at baseline for DIAGNODE-B.

Static				Previous Trial					DIAGNODE-B						
Subject	Gender	Relative with T1D	HLA	Previous Trial (Number of Previous GAD-Alum Injections)	Age (Year)	BMI (kg/m^2^)	Time Since Diagnosis (Days)	Baseline C-Peptide (nmol/L)	Baseline HbA1c (mmol/mol)	Age (Year)	BMI (kg/m^2^)	Time Since Diagnosis (Months)	Time Since the Last GAD-Alum Injection (Months)	Baseline C-Peptide (nmol/L)	Baseline HbA1c (mmol/mol)
A1	Male	Yes	DR3-DQ2/DR4-DQ7	DIAGNODE-1 (4)	21	23.43	26	0.415	66	28	27.93	76	44	0.13	70
A2	Male	Yes	DR3-DQ2/DR10-DQ5.1	DIAGNODE-1 (4)	23	19.05	41	0.375	49	29	22.70	76	43	0.02	50
B1	Female	No	DR3-DQ2/DR4-DQ8	DIAGNODE-2 (3)	13	18.30	76	0.639	36	18	23.63	54	49	0.08	43
B2	Male	No	DR3-DQ2/DR4-DQ8	DIAGNODE-2 (3)	17	20.00	145	0.755	46	22	22.05	55	48	0.01	61
B3	Male	Yes	DR3-DQ2/DR4-DQ8	DIAGNODE-2 (3)	13	18.80	144	0.540	50	17	23.61	46	39	0.03	47
B4	Female	No	DR3-DQ2/DR4-DQ8	DIAGNODE-2 (3)	23	26.60	73	1.246	45	26	29.34	44	39	0.06	47

## Data Availability

Data are available from corresponding author on reasonable request after ethical approval.

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
