# Peer review of "Redosing with Intralymphatic GAD-Alum in the Treatment of Type 1 Diabetes: The DIAGNODE-B Pilot Trial"

_ijms, 2025, doi:10.3390/ijms26010374_

Round 1
Reviewer 1 Report (New Reviewer)
Comments and Suggestions for Authors
1. I noticed that there isn’t a control group in the study, and adding one would really help clarify the results and provide a more meaningful comparison to better understand how the treatment is working.
2. The C-peptide changes over the 12 months seem small, so it would be great to tie those results to clinical outcomes to give a better sense of how the treatment is affecting patients in real-world terms.
3. While Vitamin D supplementation is included, it would be helpful to dive deeper into its potential impact on both the immune response and beta-cell function, as this could provide more context to the findings.
4. Lastly, in the conclusion, it would be really useful for the authors to discuss the long-term implications of their findings, especially in terms of how redosing could affect managing type 1 diabetes in the future.
The quality of the English language could be enhanced.
Author Response
Please see the attachment.

Reviewer 2 Report (New Reviewer)
Comments and Suggestions for Authors
Current report demonstrated that redosing with GAD-alum is safe and tolerable for patients with type-1 diabetes. Please conduct the concerns below.
1. Abbreviation must show the full spelling at first time, such as DR3DQ2.
2. Current study seems continued the previous reports. But, the potential aims were not mentioned in the introduction in clear.
3. Source of the GAD-alum and details of trials were unknown.
4. The mean C-peptide AUC in comparison without discussion. Why?
5. Limitations of small sample size were ignored.
6. Please check line 186 carefully.
7. The C-peptide decreased minimally during the 12 months trial period in type-1 diabetic patients. It needs reference(s) to support.
8. In conclusion, redosing with GAD-alum is safe and tolerable that will be better to replace “is” as “seems” in revision.
Author Response
Please see the attachment.

Reviewer 3 Report (New Reviewer)
Comments and Suggestions for Authors
The article titled "Re-dosing with intralymphatic GAD-alum in the treatment of Type 1 Diabetes. The DIAGNODE-B pilot trial." is a follow-up study of previous trial with an additional injection of the GAD-alum where patients were followed up for 12 months. The conclusion is supported by the data included in the manuscript. Please make the objective of the study clear in the abstract and introduction section because it was not clear to me what the study is about and need to go through again. The aim of the trail and the need should be clearly and separately mentioned.
The author mentioned that GADA is a marker of inflammation, but GADA is an extended marker of autoimmune profile. Please make it clear. Is the results are for autoimmune response or the presence of systemic inflammation?
The limitations of the study should be mentioned separately after the conclusion section.
Table showed the demographics. Were there any associated comorbid conditions.
Round 2
Reviewer 1 Report (New Reviewer)
Comments and Suggestions for Authors
Citing the article https://www.mdpi.com/2562486 would be a good idea because of its quality. This reference may strengthen the research, providing a solid foundation for further discussion.
Additionally, the article is ready for advancement through the publication process.
Author Response
Citing the article https://www.mdpi.com/2562486 would be a good idea because of its quality. This reference may strengthen the research, providing a solid foundation for further discussion.
Additionally, the article is ready for advancement through the publication process.
Response: Thank you for the positive feedback. We have added the suggestd refeence in the Discussion.
This manuscript is a resubmission of an earlier submission. The following is a list of the peer review reports and author responses from that submission.
Round 1
Reviewer 1 Report
Comments and Suggestions for Authors
The work aligns with the global trend of seeking molecules that would inhibit the process of autoimmune destruction of beta cells at the early stages of type 1 diabetes development. The presented study showcases the results of a project that serves as a continuation of the Diagnostics 1 and 2 studies, in which a selected group of patients from previous projects received an additional dose of GAD-alum. The small number of patients included in the study represents a limitation, as noted by the authors themselves. It can be inferred that the ongoing Diagnostics 3 study is a response to the conclusion derived from the presented study (Diagnostics B), suggesting that the tested product should be administered as early as possible in relation to the diagnosis of type 1 diabetes.
Comments: In the title, it would be beneficial to add the acronym Diagnostics B at the end, as well as in the abstract. This is necessary because the reader must infer which of the projects presented in the diagram is being discussed in the paper.
In the introduction section, it would be worthwhile to briefly refer to the immunological basis of GAD-alum's action. This is especially pertinent for readers who have not reviewed previous works on the topic. At the end of this section, it would be appropriate to note that the described project constitutes the Diagnostics B study.
In the results section, and in many other areas where descriptions are related to tables or diagrams, there is the notation "Error! Reference source not found." As a reviewer, I am left to infer which table or diagram is being referenced in the text. For example, in lines 83 and 84, one must guess that Figure 1 indeed refers to Consort Diagram 1; the actual Table 1 is found in section 2.2.
In Figure 1, the description should clarify what the terms A1, A2, B1, B2, B3, and B4 represent, although it is intuitively understood that they refer to successive patients.
In section 2.3, similar to section 2.1 (line 131), there is a lack of description indicating that the information is contained in Figure 4 (similarly, "Error! Reference source not found" appears here, as well in lines 148 and 151). In section 2.4, we find an analogous situation (lines 160, 162, 164, 166, 169, and 177). It would be inferred that this refers to the supplementary tables located at the end of the paper.
Scheme 1 is missing; it should be inferred that some of the data is included in Figure 5 (line 195).
The limitations of the study (lines 237, 238) could be delineated but this is not mandatory.
In summary, the described project is extremely interesting and worthy of publication in the IJMS, provided that the outlined comments are addressed.
Reviewer 2 Report
Comments and Suggestions for Authors
Queries/Concerns/Suggestions:
· If words limit is not the concern, please avoid abbreviation in the manuscript title.
· What is HLA DR3DQ2? Authors should briefly clarify in abstract and explain it in detail in the introduction section.
· Minimize use of abbreviations in abstract, e.g., HLA DR3DQ2, GAD, HbA1c, IDAA1c, TIR etc.
· Suggest, authors to mention epidemiological statistics details for type 1 diabetes in introduction.
· Line #83-84 “The clinical trial design is illustrated in Error! Reference source not found.. The baseline characteristics of the patients are shown in Error! Reference source not found..” Correct this and several other Errors in reference citing in entire manuscript,
· Figure 1 legends are missing.
· Table 1: too wordy, Rows & Columns widths are not aligned with the text. Hence, very difficult to read and understand the details.
· Figure numbers are not in serial, there are two figure 1 and no figure 2. Same for figure citation in the text.
· Part of Figure 4 is missing.
· What authors mean by Figure XX?
· Graphical representation of data is very poor. Authors should use graphPad prism or any other similar software to make graphs and do statistical comparisons.
· Line #195-199, what these details, mentioned as Scheme 1. Please clarify.
Comments on the Quality of English LanguageThe manuscript English in not up to the publication’s standards. Also, there are few instances of Spacing, and Grammatical errors. Hence, suggest authors to carefully check entire manuscript for the precision of English Language, Grammer, and spacing, if needed authors cloud seek for scientific English editing help.
